# I²BQ: QUANTIZING LLMS VIA INTRA- AND INTER-BLOCK OPTIMIZATION

## ABSTRACT

Post-training quantization (PTQ) has emerged as a promising solution for reducing the memory and computation overhead of large language models (LLMs), enabling efficient deployment without requiring full model retraining. However, existing PTQ methods struggle with weight–activation joint quantization and extreme weight quantization. The main challenge stems from the depth and cross-layer dependencies of LLMs, which cause quantization errors to propagate and accumulate across layers, leading to degraded performance. In this paper, we present I²BQ, a simple yet effective framework that simultaneously addresses weight–activation joint quantization and extreme weight quantization. We first propose a granular quantization strategy that treats self-attention and feed-forward (FFN) modules as separate quantization units with module-specific optimization objectives. To mitigate inter-layer error accumulation, we introduce an inter-block quantization strategy that explicitly accounts for cross-layer dependencies by encouraging consistency between blocks. Extensive experiments across diverse LLMs, including OPT and the LLaMA family, demonstrate that I²BQ achieves superior performance under both W4A4 and highly aggressive W2 settings, while incurring negligible additional computational overhead.

## 1 INTRODUCTION

Large Language Models (LLMs) have gained significant attention for their remarkable performance across a wide range of tasks Wang et al. (2019); Adiwardana et al. (2020). However, their practical deployment remains severely constrained by their immense computational and memory requirements, driven by the sheer scale of model parameters. For instance, GPT-3, with 175 billion parameters, demands hundreds of gigabytes of memory, leading to substantial energy consumption. Thus, reducing inference costs of LLMs has emerged as a critical and active area of research.

Model quantization offers a feasible solution to the inference inefficiencies of large models by converting high-precision data types (e.g., float32) into low-bit representations such as int4. This transformation can reduce the memory footprint by up to $8\times$ and substantially improve computational throughput. Among various quantization methods, post-training quantization (PTQ) is particularly appealing due to its deployment efficiency. It enables lightweight adaptation of pretrained models using only a small calibration dataset, without requiring expensive full-model retraining, thereby making it highly practical for real-world applications.

Early PTQ methods Wu et al. (2016) are primarily developed for convolutional neural networks (CNNs). These approaches typically use a small unlabeled dataset to determine appropriate scaling factors or clipping thresholds. However, directly applying such techniques to LLMs introduces new challenges. Unlike CNNs, LLMs exhibit systemic Dettmers et al. (2022) and extremely large outliers (e.g., exceeding 2000) An et al. (2025). Naïvely clipping these outliers can lead to severe degradation in accuracy as they often encode critical information for model performance. To address this issue, a variety of LLM-specific PTQ techniques have been proposed. For example, SmoothQuant Xiao et al. (2023) introduces a diagonal rescaling matrix to shift activation outliers into the weight domain, thereby simplifying the activation distribution. Quarot Ashkboos et al. (2024) utilizes Hadamard transformations to regularize activation distributions, promoting uniformity and reducing quantization error. To further enhance quantization performance, subsequent methods such as SPINQuant Liu et al. (2024) and FlatQuant Sun et al. (2024) design more sophis-

ticated transformation strategies to better handle outlier migration. However, these methods often incur additional computational overhead and complexity, and their performance remains limited, especially in extreme low-bit scenarios such as 2-bit quantization (W2).

Another major challenge in LLM quantization is the cumulative nature of quantization errors across layers. Due to their large parameter counts and deep architectures, LLMs are particularly vulnerable to error accumulation, which can result in significant performance degradation as errors propagate through successive layers. To solve this, OmniQuant Shao et al. (2023) proposes a block-wise quantization error minimization strategy, which learns quantization-specific parameters to reduce quantization-induced discrepancies. However, OmniQuant employs an indirect optimization strategy to approximate the effects of quantization on weights and activations, which may lead to suboptimal performance in certain scenarios, particularly under aggressive quantization settings.

In this paper, we first reveal the functional and distributional difference between the self-attention and feedforward network modules, as well as the inter-block dependencies in LLMs. Motivated by these observations, we present $I^2BQ$, a simple yet effective framework for weight–activation joint quantization and extreme weight quantization. Specifically, we introduce a granular quantization strategy that treats self-attention and feed-forward modules as separate quantization units, each optimized with module-specific objectives reflecting their distinct functional roles. To mitigate cumulative quantization errors, we further propose an inter-block quantization strategy that explicitly accounts for dependencies between Transformer blocks. This encourages consistency across layers and effectively reduces error propagation through the network. Our main contributions are:

- We propose treating self-attention and FFN modules within each transformer block as separate quantization units to enable finer-grained control and reduce quantization errors.

- To mitigate error accumulation across blocks during quantization, we design a cross-block error compensation mechanism that minimizes error propagation throughout the network.

- Our method consistently achieves superior quantization performance in both W4A4 and highly aggressive W2 settings, for weights and activations, across diverse LLMs including OPT and the LLaMA family, while incurring negligible additional computational overhead.

## 2 PRELIMINARIES

### 2.1 GENERAL QUANTIZATION STRATEGIES

Quantization techniques convert high-precision numerical formats into compact low-bit representations, enabling significant gains in memory efficiency and computational speed. According to the quantization target, existing quantization methods for LLMs can be categorized into weight-only quantization and joint quantization. Weight-only quantization aims to represent model weights using low-bit formats (e.g., 4-bit) while maintaining activations in full precision (typically 32-bits) Lin et al. (2024b). In contrast, joint quantization Shao et al. (2023) compresses both weights and activations to achieve higher efficiency, albeit at the cost of potentially greater quantization errors. Based on the optimization strategy, LLM quantization can be further classified into quantization-aware training (QAT) Liu et al. (2023); Chen et al. (2024) and post-training quantization (PTQ) Huang et al. (2024); Li et al. (2023). QAT involves retraining the model to learn low-precision weights under quantization constraints, while PTQ directly quantizes pretrained weights without additional retraining. In this work, we primarily focus on joint quantization with PTQ due to its practicality. This strategy requires only minimal calibration data and significantly reduces computational overhead compared to QAT.

### 2.2 BASIC QUANTIZATION PROCESS

A classical quantization approach, integer uniform quantization Jacob et al. (2018), aims to convert floating-point values into uniformly spaced integer representations. Given a floating-point input $\mathbf{F}$

(which can be a vector or matrix), its $b$-bits quantized representation $\mathbf{F}_b$ is computed as follows:

$$\mathbf{F}_b = \text{clamp}\left(\left\lfloor \frac{\mathbf{F}}{\alpha} \right\rceil + z, \ 0, \ 2^b - 1\right), \tag{1}$$

$$\alpha = \frac{\gamma \max(\mathbf{F}) - \beta \min(\mathbf{F})}{2^b - 1}, \tag{2}$$

$$z = -\left\lfloor \frac{\beta \min(\mathbf{F})}{\alpha} \right\rceil, \tag{3}$$

where $\lfloor \cdot \rceil$ denotes rounding to the nearest integer, $\gamma$ and $\beta$ are optional clipping coefficients that control the influence of extreme values. The scale factor $\alpha$ maps the range of $\mathbf{F}$ to the target integer range, while the zero-point offset $z$ aligns the minimum scaled value with zero in the quantized space.

In LLMs, the core architectural unit is the transformer block, which comprises several key components including multi-head self-attention, a feedforward network (FFN), layer normalization, and residual connections. The linear layers within self-attention and FFN modules account for the majority of memory consumption and inference latency. Consequently, most LLM quantization approaches primarily target these linear layers, while keeping non-linear operations such as `Softmax` (used in attention) and activation functions like `Swish` in full precision to preserve numerical stability and model accuracy. Specifically, for a give layer $l$, its output embedding can be expressed as $\mathbf{Y} = \mathbf{A}\mathbf{W}^\top$, where $\mathbf{A}$ and $\mathbf{W}$ are the activation and weight matrices, respectively. In this work, we adopt joint quantization, quantizing both $\mathbf{A}$ and $\mathbf{W}$ into $b_1$-bits and $b_2$-bits representations (e.g., W4A4 refers to 4-bit weights and activations).

## 2.3 ACTIVATION OUTLIERS IN LLMS

One of the most significant challenges in LLM quantization lies in the presence of activation outliers, which can severely degrade the performance of low-bit quantization methods Dettmers et al. (2022). Unlike in convolutional neural networks (CNNs), where outliers can often be clipped without notable performance loss Zhao et al. (2019), activation outliers in LLMs typically carry critical information essential for maintaining model performance. These outliers not only appear in a structured pattern but also as isolated values with extreme magnitudes, making them particularly difficult to handle in quantization. To mitigate this issue, a range of outlier-aware quantization techniques have been proposed. For example, GPT3.int8() Dettmers et al. (2022) introduces a mixed-precision group-wise quantization strategy that selectively applies higher precision to sensitive channels based on outlier detection. Smoothquant Xiao et al. (2023) mitigates quantization difficulty by shifting the burden from activations to weights via layer-wise affine transformations. Quarot Ashkboos et al. (2024) applies learnable rotation matrices to the inputs and outputs of linear layers, aligning activation and weight distributions to reduce quantization error. Subsequent extensions Liu et al. (2024); Lin et al. (2024a) explore alternative transformation schemes to promote activation uniformity. While these methods improve accuracy, they often incur substantial computational overhead, limiting their practicality.

## 2.4 CUMULATIVE ERRORS IN QUANTIZATION

In addition to addressing activation outliers, it's also crucial to optimize the cumulative quantization error propagated across layers. A representative solution is BRECQ Li et al. (2021), a method designed for CNNs that minimizes PTQ accuracy loss by performing gradient-based optimization over weights using a small calibration set to guide optimal rounding. However, directly applying BRECQ to LLMs is impractical due to the massive parameter scale (often billions), which results in an overwhelming optimization space and renders weight optimization computationally infeasible. To address this challenge, OmniQuant Shao et al. (2023) introduces a block-wise quantization error minimization strategy, which avoids optimizing all parameters and instead focuses on learnable quantization parameters (e.g., affine transformation parameters for each channel). These parameters are optimized to minimize the reconstruction error within each block. Nevertheless, OmniQuant optimizes only the quantization parameters, without directly updating the weights or activations themselves, which may lead to suboptimal quantization performance in certain cases.

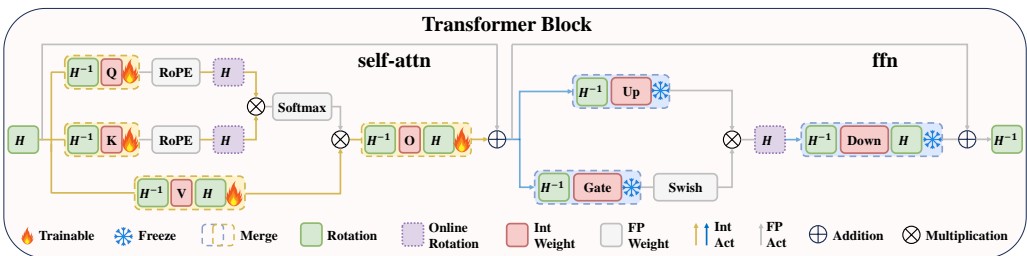

Figure 1: Overview of the I²BQ framework within a transformer block. The module-wise optimization is applied to both self-attention and feed-forward. A Hadamard-based rotation ($H$) is first applied to the input and weight matrices, followed by quantization and per-module optimization.

## 3 METHODOLOGY

### 3.1 INSIGHTS WITHIN TRANSFORMER BLOCKS

Existing LLM quantization approaches usually treat the entire transformer block as the basic unit for reconstruction, i.e., minimizing the quantization error between the outputs of the quantized and original blocks. However, this coarse-grained reconstruction strategy can lead to sub-optimal performance due to several important factors.

1 *Self-attention and FFN modules serve fundamentally different functions.* The self-attention module captures cross-token dependencies by modeling contextual relationships across the sequence, enabling global information aggregation. In contrast, the FFN module processes each token independently to enrich its representations. These distinct roles in information processing are a hallmark of the Transformer's functionally specialized design. However, computing the reconstruction loss at the level of the entire block neglects this separation of concerns, potentially undermining the specialized modeling capacity of each module and leading to sub-optimal quantization behavior.

2 *Residual connections are separately applied to self-attention and FFN.* In Transformer blocks, residual connections serve as unquantized information bypasses, helping to mitigate quantization errors in the forward pass and preserving gradient flow during backpropagation. As illustrated in Figure 1, these residual paths are constructed independently for the self-attention and FFN modules, rather than built only one for each block. This architectural design enhances robustness, modularity, and training stability. We argue that the quantization strategy should respect this modular disentanglement by applying reconstruction loss separately to each module, rather than enforcing a unified loss over the entire block. Otherwise, the gradients and error signals may become entangled across the two functionally distinct operations, thereby degrading performance.

3 *Self-attention and FFN modules exhibit significant distribution differences.* As shown in Figure 2 and Figure 3 (a) and (c), the activations from the self-attention and FFN modules exhibit distinct distribution characteristics. Even after applying rotation transformation, this discrepancy persists, as illustrated in Figures 3 (b) and (d). However, a block-level quantization strategy that treats the entire Transformer block uniformly fails to account for this distributional divergence.

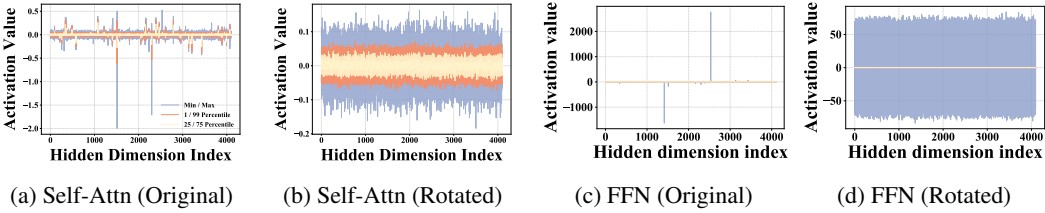

(a) Self-Attn (Original)  (b) Self-Attn (Rotated)  (c) FFN (Original)  (d) FFN (Rotated)

Figure 3: Activation distribution statistics before and after applying rotation transformation to the self-attention and FFN in Block 2 of LLaMA2-7B. The two types consistently exhibit distinct activation distributions, regardless of rotation. Each subplot visualizes the minimum/maximum, 1st/99th percentiles, and 25th/75th percentiles across hidden dimensions.

## 3.2 MODULE-WISE OPTIMIZATION

Motivated by the above empirical observation and analysis, we propose a more granular quantization strategy that independently optimizes the quantization errors for the self-attention and feed-forward modules. Specifically, we design module-specific optimization objectives tailored to the distinct functional roles of each module.

For the self-attention module, we jointly optimize the quantization parameters for linear layers, including the query, key, value, and output projections, along with their corresponding activations. Given an input $x$, we first construct a standard $L_2$ reconstruction loss between outputs of the quantized and full-precision self-attention module $f_{\text{self-attn}}$:

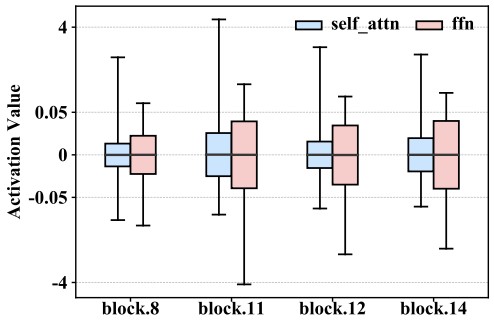

Figure 2: Box plots of activation distributions for the self-attention and FFN in selected transformer blocks of LLaMA2-7B.

$$\mathcal{L}_1 = \left\| \widetilde{f}_{\text{self-attn}}(x) - f_{\text{self-attn}}(x) \right\|_2^2 \qquad (4)$$

where $\widetilde{f}_{\text{self-attn}}$ denotes the corresponding quantized version of the self-attention module. To further preserve the structural relationships captured by attention mechanisms, we introduce an attention-preserving loss that aligns the attention maps between the quantized and full-precision models. Using Kullback–Leibler (KL) divergence, this loss encourages the quantized model to retain inter-token dependencies:

$$\mathcal{L}_2 = \sum_{i=1}^{N} \sum_{j=1}^{N} \mathbf{M}_{ij} \cdot \log\left( \frac{\mathbf{M}_{ij}}{\widetilde{\mathbf{M}}_{ij} + \varepsilon} \right) \qquad (5)$$

where $N$ is the sequence length, $\varepsilon$ is a small constant for numerical stability, $\mathbf{M} \in \mathbb{R}^{N \times N}$ and $\widetilde{\mathbf{M}} \in \mathbb{R}^{N \times N}$ represent the attention matrices computed from the full-precision and quantized query-key interactions, respectively. By minimizing this loss, the quantized attention module is guided to preserve the relational structure encoded by the original model, thus enhancing its fidelity under low-bit constraints. Then, the overall quantization loss for the self-attention module $\mathcal{L}_{\text{self-attn}}$ is formulated as a weighted combination:

$$\mathcal{L}_{\text{self-attn}} = \mathcal{L}_1 + \lambda \mathcal{L}_2 \qquad (6)$$

For the FFN module, we quantize the gate, up, and down projection layers collectively to maintain internal consistency. Analogous to the self-attention module, we construct an $L_2$ reconstruction loss to minimize the quantization error between the quantized and full-precision outputs of the FNN module:

$$\mathcal{L}_{\text{FFN}} = \left\| \widetilde{f}_{\text{FFN}}(x) - f_{\text{FFN}}(x) \right\|_2^2 \qquad (7)$$

This loss encourages accurate approximation of the original representations while preserving the FFN's token-wise transformation capability under quantization.

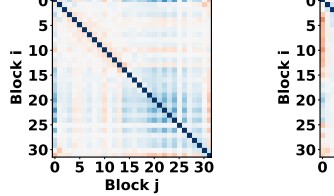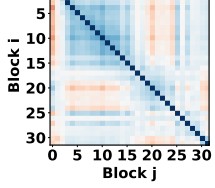

Figure 4: Visualizations of the Hessian matrices for the 32 self-attention components (left) and 32 FFN components (right) of LLaMA2-7B. The blue off-diagonal regions in the inter-component Hessian matrix indicate a strong dependency between these components.

## 3.3 INSIGHTS ACROSS TRANSFORMER BLOCKS

LLMs are built upon the Transformer architecture, which consists of a stack of Transformer blocks arranged sequentially. These blocks are often

Table 1: Comparison of perplexity on WikiText2 (↓) and average accuracy on nine zero-shot tasks (↑). FP16 denotes full precision. The best are bold-faced and second-best are underlined.

| #Bits W-A-KV | Method | LLaMA-3 8B 0-shot Wiki | | LLaMA-3 70B 0-shot Wiki | | LLaMA-2 7B 0-shot Wiki | | LLaMA-2 13B 0-shot Wiki | | LLaMA-2 70B 0-shot Wiki | | LLaMA 7B 0-shot Wiki | | LLaMA 13B 0-shot Wiki | |
|---|---|---|---|---|---|---|---|---|---|---|---|---|---|---|---|
| | | Avg.(↑) | (↓) | Avg.(↑) | (↓) | Avg.(↑) | (↓) | Avg.(↑) | (↓) | Avg.(↑) | (↓) | Avg.(↑) | (↓) | Avg.(↑) | (↓) |
| 16-16-16 | **FP16** | 68.09 | 6.14 | 73.81 | 2.86 | 65.21 | 5.47 | 67.61 | 4.88 | 71.59 | 3.32 | 64.48 | 5.68 | 66.67 | 5.09 |
| 4-16-16 | RTN | 63.70 | 8.13 | 31.15 | $1e^5$ | 61.27 | 7.02 | 60.24 | 6.39 | 69.62 | 3.87 | 62.67 | 7.94 | 63.45 | 8.60 |
| | SmoothQuant | 62.79 | 8.12 | 67.94 | 6.70 | 58.88 | 8.03 | 62.03 | 5.86 | 65.93 | 5.50 | 62.24 | 7.46 | 62.69 | 18.75 |
| | GPTQ | 61.03 | 7.43 | 31.45 | $9e^3$ | 60.86 | 9.84 | 64.71 | 5.79 | 70.96 | 3.94 | 60.15 | 7.93 | 64.36 | 6.58 |
| | OmniQuant | 65.66 | 7.19 | – | – | 63.19 | 5.74 | 66.38 | 5.02 | 71.04 | 3.47 | 63.42 | 5.86 | 66.22 | 5.21 |
| | AWQ | 67.03 | 7.36 | 68.92 | 5.92 | 63.89 | 5.83 | 66.25 | 5.07 | 70.88 | 4.03 | 63.30 | 5.97 | 65.58 | 5.28 |
| | QuaRot | 67.27 | 6.53 | 72.93 | 3.53 | **64.30** | 5.62 | 66.95 | 5.00 | 71.21 | **3.41** | 63.40 | 5.83 | 65.91 | 5.20 |
| | SpinQuant | 66.54 | 6.49 | 72.90 | 3.49 | 63.59 | **5.58** | 67.14 | 5.00 | 71.12 | 3.43 | 63.94 | **5.76** | 66.32 | **5.16** |
| | I²BQ | **67.68** | **6.46** | **73.25** | **3.28** | 64.16 | 5.59 | **67.23** | **4.98** | **71.32** | **3.41** | **64.09** | 5.79 | **66.47** | 5.18 |
| 4-4-16 | RTN | 33.42 | $6e^2$ | 31.21 | $8e^3$ | 32.44 | – | 30.86 | $8e^3$ | 30.90 | $7e^4$ | 32.51 | $7e^3$ | 31.63 | $3e^4$ |
| | SmoothQuant | 33.04 | $10^3$ | 34.67 | $2e^2$ | 32.13 | – | 34.26 | $10^3$ | 35.86 | $3e^2$ | 34.42 | $3e^2$ | 33.29 | $6e^2$ |
| | GPTQ | 32.98 | $5e^2$ | 31.47 | $4e^4$ | 32.72 | – | 30.11 | $4e^3$ | 30.86 | – | 32.12 | $10^3$ | 31.51 | $3e^3$ |
| | QuaRot | 61.69 | 8.02 | 65.56 | 6.35 | 61.87 | 6.05 | 65.13 | 5.35 | 69.96 | 3.78 | 61.76 | 6.22 | 64.46 | 5.50 |
| | SpinQuant | 64.11 | 7.28 | 66.99 | 6.10 | 57.37 | 6.78 | 63.23 | 5.24 | 70.58 | 3.68 | 61.82 | 6.08 | 64.59 | 5.36 |
| | I²BQ | **65.01** | **7.26** | **72.09** | **4.02** | **63.67** | **5.82** | **66.13** | **5.16** | **70.81** | **3.61** | **62.48** | **6.06** | **65.56** | **5.35** |
| 4-4-4 | RTN | 33.18 | $7e^2$ | 30.82 | $8e^3$ | 32.67 | – | 30.93 | $7e^3$ | 31.73 | $7e^4$ | 32.87 | $10^4$ | 31.33 | $3e^4$ |
| | SmoothQuant | 32.96 | $10^3$ | 33.76 | $3e^2$ | 32.12 | – | 33.36 | $10^3$ | 35.54 | $3e^2$ | 33.32 | $3e^2$ | 33.28 | $5e^2$ |
| | GPTQ | 33.71 | $6e^2$ | 31.20 | $4e^4$ | 33.52 | – | 27.85 | $5e^2$ | 31.09 | – | 31.80 | $2e^3$ | 30.63 | $3e^3$ |
| | OmniQuant | 32.33 | $4e^2$ | – | – | 48.40 | 14.26 | 50.35 | 12.30 | – | – | 48.46 | 11.26 | 45.63 | 10.87 |
| | QuaRot | 61.38 | 8.18 | 65.33 | 6.60 | 61.48 | 6.11 | 65.16 | 5.39 | 70.30 | 3.80 | 61.22 | 6.26 | 64.59 | 5.53 |
| | SpinQuant | 64.10 | 7.35 | 66.31 | 6.24 | 62.01 | 5.96 | 64.13 | 5.74 | 70.57 | 3.61 | 61.32 | 6.12 | 64.95 | 5.39 |
| | I²BQ | **65.07** | **7.33** | **71.33** | **4.41** | **63.00** | **5.96** | **65.21** | **5.24** | **70.68** | **3.59** | **62.12** | **6.08** | **65.21** | **5.38** |

tightly coupled, exhibiting strong representational dependencies across their layers. Due to the inherently sequential and compositional structure of Transformer, quantization errors introduced in one block can propagate through the network, potentially impacting not only that block but also the subsequent ones. To investigate this inter-block dependency, we evaluate the similarities structure across different transformer blocks. Figure 4 illustrates the Hessian matrix computed across 32 consecutive transformer blocks, including both self-attention and FFN components. As shown, several off-diagonal entries are notably non-zero, particularly among FFN modules, which indicates the presence of second-order dependencies between different blocks. These inter-block relationships imply that quantization-induced information loss in one block may be captured or amplified by subsequent blocks. Therefore, when designing quantization strategies, it is important to consider not only the local reconstruction loss within a block but also its downstream impact on later blocks. Incorporating such cross-block effects into the optimization process can lead to more robust and globally consistent quantization.

## 3.4 Cross-Block Error Compensation

Building on the above analysis and empirical observations, we propose a cross-block optimization approach to account for inter-block dependencies during quantization. As illustrated in Figure 5 (b), most existing quantization methods operate in a block-wise manner, quantizing each Transformer block independently while ignoring the representational dependencies across blocks. To address this limitation, we introduce an inter-block quantization strategy (Figure 5 (c)), which promotes consistency across sequential blocks and helps mitigate the propagation of quantization errors through the network.

Specifically, for a give input $x$, define the following loss function that measures the discrepancy between the full-precision and quantized outputs over a sequence of blocks from index $i$

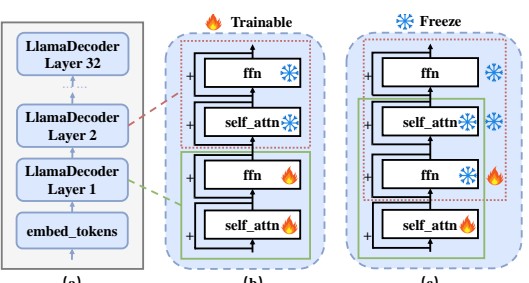

Figure 5: Illustration of cross-block error compensation. (a) Structure of LLaMA2-7B; (b) existing methods optimize quantization error separately within each decoder layer; (c) our method performs cross-block optimization to minimize cumulative quantization error.

Table 2: Perplexity on Wiki and C4 for various quantization methods across LLaMA-1 and LLaMA-2 models. The best results are bold-faced and the second-best results are underlined.

| #Bits W-A-KV | Method | LLaMA2-7B | | LLaMA2-13B | | LLaMA2-70B | | LLaMA1-7B | | LLaMA1-13B | | LLaMA1-30B | |
|---|---|---|---|---|---|---|---|---|---|---|---|---|---|
| | | Wiki(↓) | C4(↓) | Wiki(↓) | C4(↓) | Wiki(↓) | C4(↓) | Wiki(↓) | C4(↓) | Wiki(↓) | C4(↓) | Wiki(↓) | C4(↓) |
| 16-16-16 | FP | 5.47 | 6.97 | 4.88 | 6.46 | 3.32 | 5.52 | 5.68 | 7.08 | 5.09 | 6.61 | 4.10 | 5.98 |
| 3-16-16 | RTN | 539.48 | 402.35 | 10.68 | 12.51 | 7.52 | 10.02 | 25.73 | 28.26 | 11.39 | 13.22 | 14.95 | 28.66 |
| | GPTQ | 8.37 | 9.81 | 6.44 | 8.02 | 4.82 | 6.57 | 8.06 | 9.49 | 6.76 | 8.16 | 5.84 | 7.29 |
| | AWQ | 24.00 | 23.85 | 10.45 | 13.07 | – | – | 11.88 | 13.26 | 7.45 | 9.13 | 10.07 | 12.67 |
| | OmniQuant | 6.58 | 8.65 | 5.58 | **7.44** | 3.92 | 6.06 | 6.49 | 8.19 | 5.68 | 7.32 | 4.74 | 6.57 |
| | QuaRot | 6.09 | 8.69 | 5.37 | 7.70 | 3.71 | 6.12 | 6.25 | 8.46 | 5.47 | 7.48 | 4.60 | 6.69 |
| | I²BQ | **5.89** | **7.82** | **5.25** | 7.49 | **3.67** | **6.01** | **6.01** | **8.03** | **5.39** | **7.24** | **4.47** | **6.48** |
| 2-16-16 | RTN | 4e⁴ | 5e⁴ | 5e⁴ | 7e⁴ | 2e⁴ | 2e⁴ | 1e⁵ | 1e⁵ | 7e⁴ | 5e⁴ | 2e⁴ | 2e⁴ |
| | GPTQ | 7e³ | – | 2e³ | 323.12 | 77.95 | 48.82 | 2e³ | 689.13 | 5e³ | 2e³ | 169.80 |
| | OmniQuant | 37.37 | 90.64 | 17.21 | 26.76 | 7.81 | 12.28 | 15.47 | 24.89 | 13.21 | 18.31 | 8.71 | 13.89 |
| | QuaRot | 22.07 | 49.68 | 12.52 | 26.58 | 6.00 | 10.50 | 12.25 | 22.65 | 9.63 | 16.22 | 7.89 | 14.17 |
| | I²BQ | **14.23** | **19.63** | **9.36** | **13.87** | **4.92** | **7.61** | **10.28** | **13.89** | **7.99** | **10.67** | **6.43** | **9.30** |

to $i + n$:

$$\min \left\| f_{i+n} \circ \cdots \circ \widetilde{f}_i(x) - f_{i+n} \circ \cdots \circ f_i(x) \right\|_2^2 \tag{8}$$

Here, $f_i$ denotes the full-precision operation of the $i$-th block, and $\widetilde{f}_i$ denotes its quantized counterpart. In practice, this loss can be applied within a specific module (e.g., self-attention or FFN) by computing the reconstruction error from the module in block $i$ to the corresponding module in block $i+n$. This approach encourages the quantized representation at earlier layers to remain aligned with downstream full-precision computations, thus improving overall quantization fidelity.

## 4 EXPERIMENTS

### 4.1 EXPERIMENTAL SETUP

**Baseline.** I²BQ is a flexible and generalizable quantization framework that supports arbitrary precision configurations. To comprehensively evaluate its effectiveness across diverse scenarios, we conduct experiments under a broad spectrum of bit-width settings, including both standard and challenging low-bit regimes: W4A16KV16, W4A4KV16, W4A4KV4, W3A16KV16, W2A16A16, and W4A8A16. For comparison, we benchmark I²BQ against a range of state-of-the-art quantization methods, including SmoothQuant Xiao et al. (2023), GPTQ Frantar et al. (2022), OmniQuant Shao et al. (2023), AWQ Lin et al. (2024b), QuaRot Ashkboos et al. (2024), SpinQuant Liu et al. (2024), and CBQ Ding et al. (2023).

**Models.** We evaluate I²BQ on a suite of representative LLM models, covering multiple scales of LLaMA (7B, 13B, 30B), LLaMA-2 (7B, 13B, 70B), LLaMA-3 (8B, 70B), and OPT (30B, 66B).

**Datasets.** Following standard protocols from prior work Shao et al. (2023); Lin et al. (2024c), we evaluate quantized model performance on both language modeling and zero-shot reasoning tasks. Specifically, perplexity is measured on WikiText2 Merity et al. (2016) and C4 Dodge et al. (2021), using a context length of 2048 tokens. For zero-shot evaluation, we use nine benchmark tasks: BoolQ Clark et al. (2019), LAMBADA Radford et al. (2019), OpenBookQA Mihaylov et al. (2018), Social IQA (SIQA) Sap et al. (2019), PIQA Bisk et al. (2020), ARC (Challenge and Easy) Clark et al. (2018), HellaSwag Zellers et al. (2019), and WinoGrande Sakaguchi et al. (2021).

**Quantization Settings.** We initialize quantization parameters using grid search on 8 samples from the Pile dataset Gao et al. (2020), each with a sequence length of 1024 tokens. Optimization is then performed on 512 samples from the Pile, also with 1024-token contexts. The learning rate for quantization parameters is set to 5e-5 by default and reduced to 2e-5 for larger models (LLaMA-1-70B, LLaMA-2-70B, and LLaMA-3-70B). We use a batch size of 4 and train for 10 epochs for W4A4 precision and 5 epochs for W2A16. The loss balancing coefficient $\lambda$ is set to 10 throughout.

### 4.2 VALIDATION ON 4-BIT SETTING

Table 1 provides a comparative evaluation of various PTQ methods across multiple LLaMA model variants. Among these methods, I²BQ consistently ranks first or second in performance across all

Table 3: Evaluation of quantization on generation datasets with perplexity (↓). Following the quantization settings of the comparison methods, we employ group quantization with a group size of 128 to quantize the weights. The best results are bold-faced and the second-best results are underlined.

| #Bits | Methods | OPT-30B | | OPT-66B | | LLaMA1-30B | | LLaMA1-65B | |
| | | Wiki | C4 | Wiki | C4 | Wiki | C4 | Wiki | C4 |
|---|---|---|---|---|---|---|---|---|---|
| W16A16 | FP | 9.56 | 10.69 | 9.34 | 10.28 | 4.10 | 5.98 | 3.53 | 5.62 |
| W4A16 | GPTQ | 9.63 | 10.80 | 9.55 | 10.50 | 4.34 | 6.16 | 3.77 | 5.77 |
| | OmniQuant | 9.71 | 10.80 | **9.37** | 10.63 | 4.19 | 6.06 | 3.62 | 5.68 |
| | CBQ | 9.65 | 10.73 | 9.41 | **10.31** | 4.14 | 6.03 | 3.59 | **5.62** |
| | I$^2$BQ | **9.59** | **10.72** | **9.37** | 10.32 | **4.13** | **6.02** | **3.40** | **5.62** |
| W2A16 | GPTQ | 9.1e3 | 1.64e4 | 6.3e3 | 4.3e3 | 1.3e4 | 7.2e3 | 1.1e4 | 8.8e3 |
| | OmniQuant | 11.00 | 12.80 | 10.59 | 12.13 | 7.14 | 9.02 | 6.01 | 7.78 |
| | CBQ | 10.51 | 12.01 | 10.25 | 11.19 | 5.58 | 7.65 | 5.25 | 7.42 |
| | I$^2$BQ | **9.99** | **11.58** | **9.82** | **11.01** | **4.87** | **6.89** | **4.74** | **7.21** |
| W4A8 | OmniQuant | 9.95 | 10.96 | 9.52 | 10.73 | 4.58 | 6.45 | 3.96 | 6.12 |
| | RPTQ | 10.22 | 11.01 | 9.46 | 10.57 | - | - | - | - |
| | CBQ | 9.83 | 10.86 | **9.44** | 10.42 | 4.32 | 6.25 | 3.84 | 5.96 |
| | I$^2$BQ | **9.64** | **10.79** | 9.45 | **10.31** | **4.29** | **6.17** | **3.71** | **5.87** |
| W4A4 | OmniQuant | 10.60 | 11.89 | 10.29 | 11.35 | 10.33 | 12.49 | 9.17 | 11.28 |
| | QLLM | – | – | – | – | 8.37 | 11.51 | 6.87 | 8.89 |
| | CBQ | 10.34 | 11.79 | 9.45 | 11.02 | 7.96 | 9.73 | 5.89 | 7.52 |
| | I$^2$BQ | **10.14** | **11.29** | **9.43** | **10.93** | **7.71** | **9.60** | **5.26** | **7.13** |

Table 4: Ablation study of I$^2$BQ's main components on LLaMA-2-7B under W2A16. ↓ is better for perplexity (WikiText-2, C4), ↑ is better for downstream accuracy.

| MWO | CBEC | BWQ | Wiki(↓) | C4(↓) | ARC-C | ARC-E | HellaSwag | LAMBADA | PIQA | Winogrande | Avg(↑) |
|---|---|---|---|---|---|---|---|---|---|---|---|
| | | | 75950 | 59636 | 21.76 | 26.18 | 25.68 | 1.02 | 52.50 | 51.46 | 29.77 |
| ✓ | | | 14.83 | 20.22 | 26.88 | 45.62 | 64.20 | 44.51 | 64.20 | 59.98 | 50.90 |
| | | ✓ | 16.86 | 22.34 | 25.43 | 56.02 | 40.32 | 31.16 | 66.38 | 54.22 | 45.59 |
| ✓ | ✓ | | 14.23 | 19.63 | 30.12 | 61.20 | 63.05 | 37.42 | 70.40 | 60.14 | 53.72 |

models. Under the relatively mild quantization setting of W4A16A16, methods like QuaRot and SpinQuant occasionally achieve slightly better results. However, as quantization becomes more aggressive—particularly in configurations like W4A4KV16 and W4A4KV4—I$^2$BQ consistently delivers the lowest perplexity on WikiText2 and superior zero-shot reasoning performance across nine benchmark tasks. These improvements are especially pronounced on larger models such as LLaMA-3 70B, demonstrating the robustness of I$^2$BQ under more challenging low-bit conditions.

### 4.3 VALIDATION ON EXTREME LOW-BIT SETTINGS

To further assess the robustness of I$^2$BQ under extreme quantization, we evaluate its performance in ultra-low-bit scenarios with 2-bit and 3-bit weights (Table 2). Even under severe compression, I$^2$BQ delivers state-of-the-art performance across most settings, notably outperforming others in the W2 configuration where alternatives degrade significantly. These findings highlight I$^2$BQ's ability to preserve model fidelity even under highly constrained precision.

### 4.4 COMPARISON WITH CROSS-BLOCK QUANTIZATION METHODS.

We further benchmark I$^2$BQ against strong baselines, with a particular focus on CBQ—a recent cross-block quantization method that jointly quantizes multiple transformer layers. Our evaluation includes several large-scale models, notably OPT-30B, OPT-66B, LLaMA-1 30B, and LLaMA-1 65B. As shown in Table 3, I$^2$BQ consistently outperforms CBQ across nearly all settings and datasets. These consistent performance gains highlight the robustness of I$^2$BQ under low-bit con-

straints. Moreover, the results validate the effectiveness of the proposed module-wise optimization strategy and cross-block error compensation in achieving accurate and reliable quantization in large-scale models.

### 4.5 ABLATION STUDIES

We conduct ablation studies to validate the contribution of each component in the I$^2$BQ framework (Table 4). Starting from a baseline using standard RTN quantization, we observe severe degradation under W2A16, highlighting the challenge of ultra-low-bit quantization. Introducing our Module-Wise Optimization (MWO) significantly improves performance, reducing WikiText2 perplexity to 14.83 and enhancing downstream accuracy. To disentangle the effect of optimization granularity, we also evaluate the Block-wise Quantization Error Minimization (BWQ) module. While BWQ reduces perplexity to 16.86, MWO achieves a greater improvement, demonstrating the advantage of designing module-wise reconstruction losses for self-attention and FFN. Finally, incorporating Cross-Block Error Compensation (CBEC) alongside MWO yields the best overall performance, with the lowest perplexity and highest accuracies across nearly all tasks. These results confirm the effectiveness of our full framework in mitigating quantization errors and maintaining performance in extreme low-bit regimes.

### 4.6 OVERHEAD ANALYSIS

Our framework introduces negligible computational overhead during inference. Prior to quantization, a Hadamard-based rotation is applied to both weights and activations. The rotation of weight matrices is performed offline and fused into the model through direct weight manipulation, thereby incurring no additional runtime cost.

For activations, the rotation is applied online during the forward pass. This operation remains highly efficient, as the Hadamard matrix contains only binary values ($\pm 1$), enabling the transformation to be implemented via simple sign flips without requiring multiplications. As a result, the runtime overhead is minimal in practice. Moreover, our quantization framework does not rely on any specialized hardware, ensuring broad compatibility and ease of deployment.

## 5 CONCLUSION

To enable efficient deployment of large language models (LLMs), we present I$^2$BQ, a novel post-training quantization framework. Existing methods typically address either weight–activation quantization or extreme weight quantization, but they often overlook the cumulative error propagation in deep Transformer architectures. In contrast, our framework effectively handles both weight–activation quantization and extreme weight quantization.

Our framework consists of two core components. First, we propose module-wise optimization, which independently quantizes self-attention and feed-forward modules using tailored reconstruction objectives that account for their distinct computational roles and activation distributions. Second, we introduce a cross-block error compensation mechanism that mitigates inter-layer quantization drift by enforcing consistency across Transformer blocks.

Extensive experiments across various LLMs demonstrate that I$^2$BQ significantly improves perplexity and downstream task performance under aggressive low-bit settings (e.g., W2A16), while incurring negligible inference overhead.

## 6 LIMITATIONS

Although I$^2$BQ demonstrates strong performance in post-training quantization (PTQ) of LLMs, several limitations remain. Under extremely low-bit settings, its accuracy still lags behind that of Quantization-Aware Training (QAT) methods. Moreover, the current optimization process is time-consuming, often taking several hours to complete. In future work, we plan to substantially reduce this optimization time while further enhancing the effectiveness and scalability of PTQ.

## STATEMENTS

### ETHICS STATEMENT

In this work, we propose a post-training quantization method for large language models (LLMs) with the aim of improving their efficiency. All experiments are conducted on publicly available datasets that do not contain personally identifiable information. We have carefully followed the ethical guidelines and submission policies of ICLR and affirm that this work complies with all applicable standards.

### REPRODUCIBILITY STATEMENT

We follow the ICLR reproducibility guidelines and ensure that our work can be reproduced. All datasets used in our experiments are publicly available. Detailed descriptions of the quantization settings and hyperparameters are provided in the main text and Appendix. We will release our code upon acceptance of the paper.

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

# A APPENDIX

## A.1 MORE RESULTS

This section offers a comprehensive presentation (Table 5-11) of results across various datasets, providing supplementary details to the Table 1.

| Model | #Bits W-A-KV | Method | ARC-c | ARC-e | BoolQ | HellaS. | Lam. | OBQA | PIQA | SIQA | WinoG. | Avg. |
|---|---|---|---|---|---|---|---|---|---|---|---|---|
| | 16-16-16 | Full Precision | 46.42 | 74.33 | 77.71 | 75.94 | 73.69 | 44.20 | 79.16 | 45.91 | 69.53 | 65.21 |
| **2-7B** | 4-16-16 | RTN | 42.15 | 67.59 | 73.06 | 72.34 | 67.18 | 41.80 | 76.50 | 44.11 | 66.69 | 61.27 |
| | | SmoothQuant | 39.59 | 65.19 | 69.82 | 68.84 | 62.27 | 40.20 | 75.95 | 44.17 | 63.85 | 58.88 |
| | | GPTQ | 42.49 | 69.53 | 61.31 | 73.83 | 67.61 | 42.40 | 77.64 | 44.52 | 68.43 | 60.86 |
| | | Omniquant | 42.49 | 71.00 | 74.34 | 73.85 | 70.70 | 44.00 | 78.40 | 44.93 | 68.82 | 63.19 |
| | | AWQ | 44.11 | 70.75 | 78.07 | 74.98 | 70.68 | 43.80 | 78.13 | 45.14 | 69.38 | 63.89 |
| | | QuaRot | 43.94 | 73.15 | 76.97 | 74.87 | 78.24 | 45.09 | 78.24 | 45.09 | 69.38 | 64.30 |
| | | SpinQuant | 43.34 | 72.69 | 73.36 | 75.10 | 73.80 | 43.00 | 77.86 | 45.60 | 67.56 | 63.59 |
| | | I²BQ | 44.37 | 74.86 | 75.26 | 75.19 | 71.81 | 43.90 | 78.62 | 45.64 | 67.83 | 64.16 |
| **2-7B** | 4-4-16 | RTN | 25.34 | 28.03 | 50.52 | 27.71 | 1.01 | 26.20 | 50.82 | 33.93 | 48.38 | 32.44 |
| | | SmoothQuant | 28.33 | 26.39 | 49.39 | 27.28 | 1.18 | 23.40 | 48.00 | 33.62 | 50.75 | 32.13 |
| | | GPTQ | 24.40 | 28.70 | 51.62 | 28.66 | 1.36 | 24.60 | 51.14 | 34.49 | 49.49 | 32.72 |
| | | QuaRot | 42.32 | 69.65 | 74.77 | 72.91 | 70.81 | 39.80 | 77.20 | 43.55 | 65.82 | 61.87 |
| | | SpinQuant | 37.54 | 62.58 | 71.16 | 70.48 | 67.16 | 34.80 | 75.46 | 39.76 | 60.62 | 57.37 |
| | | I²BQ | 43.89 | 74.36 | 73.62 | 75.01 | 72.24 | 42.80 | 77.95 | 45.02 | 67.98 | 63.67 |
| **2-7B** | 4-4-4 | RTN | 27.22 | 27.06 | 50.83 | 27.34 | 0.93 | 25.80 | 49.51 | 34.85 | 50.51 | 32.67 |
| | | SmoothQuant | 26.37 | 25.63 | 47.71 | 27.05 | 1.11 | 26.40 | 51.90 | 34.49 | 48.38 | 32.12 |
| | | GPTQ | 26.96 | 27.65 | 52.84 | 28.83 | 1.63 | 29.20 | 49.62 | 35.11 | 49.80 | 33.52 |
| | | Omniquant | 31.40 | 53.75 | 63.79 | 55.06 | 35.63 | 34.40 | 66.59 | 40.28 | 54.70 | 48.40 |
| | | QuaRot | 41.43 | 69.32 | 74.19 | 72.50 | 70.66 | 39.80 | 77.42 | 43.35 | 64.64 | 61.48 |
| | | SpinQuant | 40.44 | 71.08 | 74.40 | 73.51 | 70.66 | 41.80 | 76.88 | 43.50 | 65.82 | 62.01 |
| | | I²BQ | 40.87 | 74.07 | 74.89 | 74.81 | 70.67 | 43.89 | 76.06 | 44.79 | 67.01 | 63.00 |

Table 5: Zero-shot commonsense question answering accuracy (↑) of LLaMA2-7B using different quantization methods and bit-width configurations across multiple datasets.

| Model | #Bits W-A-KV | Method | ARC-c | ARC-e | BoolQ | HellaS. | Lam. | OBQA | PIQA | SIQA | WinoG. | Avg. |
|---|---|---|---|---|---|---|---|---|---|---|---|---|
| | 16-16-16 | Full Precision | 49.15 | 77.53 | 80.58 | 79.39 | 76.62 | 45.20 | 80.63 | 47.49 | 71.90 | 67.61 |
| **2-13B** | 4-16-16 | RTN | 42.92 | 66.54 | 71.38 | 66.62 | 68.99 | 39.40 | 76.93 | 44.06 | 65.35 | 60.24 |
| | | SmoothQuant | 46.25 | 70.45 | 74.92 | 69.16 | 70.49 | 39.80 | 77.86 | 45.14 | 64.17 | 62.03 |
| | | GPTQ | 49.63 | 73.95 | 74.83 | 73.77 | 73.20 | 42.40 | 78.51 | 45.50 | 70.64 | 64.71 |
| | | Omniquant | 48.29 | 75.42 | 77.92 | 77.80 | 75.59 | 45.20 | 80.41 | 46.62 | 70.17 | 66.38 |
| | | AWQ | 48.63 | 78.16 | 78.81 | 78.48 | 75.20 | 45.00 | 79.54 | 46.21 | 72.45 | 66.25 |
| | | QuaRot | 49.15 | 76.26 | 80.46 | 78.17 | 76.50 | 45.40 | 80.03 | 45.50 | 71.11 | 66.95 |
| | | SpinQuant | 49.15 | 77.48 | 79.27 | 78.46 | 77.10 | 44.60 | 80.03 | 46.47 | 71.67 | 67.14 |
| | | I²BQ | 49.20 | 76.87 | 80.52 | 78.16 | 76.62 | 44.90 | 80.16 | 46.99 | 71.67 | 67.23 |
| **2-13B** | 4-4-16 | RTN | 27.99 | 26.81 | 38.50 | 26.08 | 0.00 | 23.60 | 48.20 | 34.90 | 51.62 | 30.86 |
| | | SmoothQuant | 24.49 | 35.06 | 47.98 | 30.87 | 3.67 | 26.20 | 55.01 | 35.31 | 49.72 | 34.26 |
| | | GPTQ | 27.82 | 26.77 | 37.92 | 25.67 | 0.00 | 21.80 | 47.77 | 35.11 | 48.15 | 30.11 |
| | | QuaRot | 46.42 | 73.86 | 78.10 | 75.68 | 74.31 | 43.00 | 79.05 | 44.37 | 71.35 | 65.13 |
| | | SpinQuant | 43.77 | 69.99 | 76.57 | 74.63 | 72.81 | 41.60 | 77.20 | 44.27 | 68.19 | 63.23 |
| | | I²BQ | 47.44 | 74.88 | 79.79 | 76.94 | 75.36 | 43.90 | 79.34 | 46.21 | 71.28 | 66.13 |
| **2-13B** | 4-4-4 | RTN | 27.82 | 26.52 | 38.38 | 26.27 | 0.02 | 26.00 | 49.78 | 34.39 | 49.17 | 30.93 |
| | | SmoothQuant | 24.49 | 33.00 | 45.84 | 30.70 | 2.70 | 23.80 | 53.81 | 34.80 | 51.07 | 33.36 |
| | | GPTQ | 27.90 | 26.39 | 37.95 | 26.16 | 0.00 | 27.00 | 48.26 | 34.39 | 50.43 | 27.85 |
| | | Omniquant | 32.85 | 55.13 | 64.34 | 60.13 | 42.85 | 33.40 | 68.17 | 39.76 | 56.51 | 50.35 |
| | | QuaRot | 47.27 | 73.91 | 78.41 | 75.33 | 73.53 | 43.80 | 79.27 | 45.85 | 69.06 | 65.16 |
| | | SpinQuant | 46.67 | 74.49 | 76.76 | 75.22 | 72.19 | 42.40 | 78.29 | 43.45 | 67.72 | 64.13 |
| | | I²BQ | 47.26 | 74.68 | 78.32 | 75.91 | 74.28 | 44.10 | 79.02 | 45.38 | 67.94 | 65.21 |

Table 6: Zero-shot commonsense question answering accuracy (↑) of LLaMA2-13B using different quantization methods and bit-width configurations across multiple datasets.

| Model | #Bits W-A-KV | Method | ARC-c | ARC-e | BoolQ | HellaS. | LambA. | OBQA | PIQA | SIQA | WinoG. | Avg. |
|---|---|---|---|---|---|---|---|---|---|---|---|---|
| | 16-16-16 | Full Precision | 57.42 | 81.02 | 83.79 | 83.81 | 79.60 | 48.80 | 82.70 | 49.18 | 77.98 | 71.59 |
| 2-70B | 4-16-16 | RTN | 55.80 | 79.29 | 81.35 | 81.78 | 75.51 | 47.60 | 81.94 | 46.83 | 76.48 | 69.62 |
| | | SmoothQuant | 50.26 | 76.56 | 81.53 | 67.81 | 73.63 | 44.40 | 81.34 | 44.17 | 73.64 | 65.93 |
| | | GPTQ | 56.91 | 80.81 | 83.24 | 82.47 | 79.06 | 47.80 | 82.75 | 48.06 | 77.51 | 70.96 |
| | | Omniquant | 57.08 | 80.81 | 82.69 | 83.07 | 79.18 | 47.40 | 83.08 | 48.87 | 77.19 | 71.04 |
| | | AWQ | 56.67 | 80.54 | 82.98 | 82.54 | 78.83 | 47.67 | 82.97 | 48.12 | 77.62 | 70.88 |
| | | QuaRot | 57.34 | 80.85 | 83.24 | 83.27 | 80.38 | 47.60 | 82.21 | 48.62 | 77.35 | 71.21 |
| | | SpinQuant | 56.91 | 80.60 | 83.18 | 83.06 | 79.16 | 49.00 | 82.75 | 48.31 | 77.11 | 71.12 |
| | | I²BQ | 57.29 | 80.97 | 83.13 | 83.02 | 80.09 | 48.80 | 82.71 | 48.62 | 77.35 | 71.32 |
| 2-70B | 4-4-16 | RTN | 29.35 | 26.05 | 37.74 | 25.97 | 0.02 | 24.80 | 51.31 | 34.14 | 48.70 | 30.90 |
| | | SmoothQuant | 25.00 | 35.98 | 55.23 | 32.52 | 7.49 | 25.00 | 54.62 | 35.21 | 51.70 | 35.86 |
| | | GPTQ | 27.82 | 25.80 | 37.95 | 25.82 | 0.00 | 27.00 | 49.67 | 33.98 | 49.72 | 30.86 |
| | | QuaRot | 55.29 | 80.35 | 81.10 | 81.87 | 79.06 | 45.80 | 82.05 | 47.90 | 76.24 | 69.96 |
| | | SpinQuant | 55.38 | 78.96 | 83.36 | 82.54 | 79.00 | 47.80 | 82.10 | 48.67 | 77.43 | 70.58 |
| | | I²BQ | 56.03 | 80.39 | 83.18 | 82.41 | 79.14 | 47.70 | 82.76 | 48.62 | 77.03 | 70.81 |
| 2-70B | 4-4-4 | RTN | 30.38 | 27.74 | 38.23 | 26.12 | 0.02 | 24.60 | 51.74 | 34.29 | 52.49 | 31.73 |
| | | SmoothQuant | 24.15 | 33.88 | 55.32 | 31.75 | 7.14 | 26.40 | 54.95 | 34.14 | 52.17 | 35.54 |
| | | GPTQ | 28.75 | 26.39 | 37.86 | 25.96 | 0.00 | 26.40 | 50.00 | 34.44 | 50.04 | 31.09 |
| | | QuaRot | 56.48 | 80.56 | 81.59 | 81.93 | 79.16 | 46.00 | 82.21 | 48.00 | 76.80 | 70.30 |
| | | SpinQuant | 56.31 | 80.64 | 83.55 | 82.36 | 79.41 | 47.20 | 82.21 | 47.29 | 76.16 | 70.57 |
| | | I²BQ | 56.31 | 80.53 | 83.33 | 82.20 | 79.09 | 47.60 | 82.32 | 48.02 | 76.71 | 70.68 |

Table 7: Zero-shot commonsense question answering accuracy (↑) of LLaMA2-70B using different quantization methods and bit-width configurations across multiple datasets.

| Model | #Bits W-A-KV | Method | ARC-c | ARC-e | BoolQ | HellaS. | LambA. | OBQA | PIQA | SIQA | WinoG. | Avg. |
|---|---|---|---|---|---|---|---|---|---|---|---|---|
| | 16-16-16 | Full Precision | 53.50 | 77.74 | 81.10 | 79.18 | 75.74 | 44.80 | 80.63 | 47.08 | 73.01 | 68.09 |
| 3-8B | 4-16-16 | RTN | 48.98 | 73.23 | 72.75 | 75.90 | 63.85 | 43.20 | 78.40 | 43.81 | 73.16 | 63.70 |
| | | SmoothQuant | 47.44 | 72.35 | 72.11 | 74.92 | 62.41 | 43.00 | 77.69 | 43.91 | 71.27 | 62.79 |
| | | GPTQ | 49.74 | 72.52 | 71.28 | 68.34 | 46.69 | 43.60 | 78.78 | 46.47 | 71.82 | 61.03 |
| | | Omniquant | 50.09 | 74.54 | 79.15 | 76.92 | 70.31 | 43.80 | 79.54 | 44.52 | 71.74 | 65.66 |
| | | AWQ | 52.22 | 76.68 | 80.31 | 77.51 | 74.81 | 44.20 | 80.14 | 46.26 | 71.67 | 67.03 |
| | | QuaRot | 51.88 | 77.53 | 79.60 | 77.87 | 73.76 | 44.80 | 79.98 | 46.37 | 73.56 | 67.27 |
| | | SpinQuant | 52.13 | 72.28 | 79.20 | 78.40 | 73.76 | 44.80 | 79.98 | 45.50 | 72.77 | 66.54 |
| | | I²BQ | 52.98 | 78.96 | 80.47 | 78.02 | 75.18 | 42.84 | 80.41 | 46.57 | 73.71 | 67.68 |
| 3-8B | 4-4-16 | RTN | 23.72 | 30.89 | 46.30 | 31.26 | 3.03 | 27.60 | 52.72 | 35.26 | 50.04 | 33.42 |
| | | SmoothQuant | 23.29 | 28.28 | 48.93 | 29.19 | 1.57 | 28.60 | 54.46 | 33.37 | 49.64 | 33.04 |
| | | GPTQ | 23.46 | 32.07 | 43.79 | 30.10 | 2.41 | 28.00 | 53.97 | 34.14 | 48.86 | 32.98 |
| | | QuaRot | 42.66 | 67.26 | 73.73 | 73.60 | 67.42 | 43.00 | 76.61 | 45.04 | 65.90 | 61.69 |
| | | SpinQuant | 47.35 | 74.12 | 76.36 | 75.98 | 69.88 | 42.46 | 77.37 | 44.47 | 68.98 | 64.11 |
| | | I²BQ | 47.97 | 74.02 | 78.66 | 76.70 | 70.77 | 43.00 | 79.56 | 45.52 | 68.90 | 65.01 |
| 3-8B | 4-4-4 | RTN | 23.72 | 30.56 | 46.18 | 29.83 | 2.70 | 28.60 | 52.45 | 34.39 | 50.20 | 33.18 |
| | | SmoothQuant | 23.55 | 28.96 | 48.84 | 28.90 | 1.44 | 29.40 | 51.09 | 34.14 | 50.36 | 32.96 |
| | | GPTQ | 23.38 | 32.74 | 44.34 | 29.72 | 2.39 | 29.80 | 54.95 | 34.75 | 51.30 | 33.71 |
| | | Omniquant | 22.87 | 30.35 | 41.53 | 31.11 | 1.86 | 25.40 | 53.37 | 34.08 | 50.43 | 32.33 |
| | | QuaRot | 42.83 | 67.42 | 73.21 | 72.66 | 66.93 | 42.20 | 75.73 | 45.19 | 66.22 | 61.38 |
| | | SpinQuant | 46.33 | 73.57 | 76.15 | 75.43 | 71.40 | 41.40 | 79.16 | 44.68 | 68.75 | 64.10 |
| | | I²BQ | 48.09 | 74.20 | 78.36 | 76.28 | 71.86 | 43.10 | 79.16 | 45.64 | 68.96 | 65.07 |

Table 8: Zero-shot commonsense question answering accuracy (↑) of LLaMA3-8B using different quantization methods and bit-width configurations across multiple datasets.

## A.2 ADDITIONAL ABLATION STUDY

Table 12 presents additional ablation study results for LLaMA2-7B under W4A4 quantization, further demonstrating the effectiveness of each module in our approach.

## A.3 HYPERPARAMETER SENSITIVITY ANALYSIS

A hyperparameter sensitivity analysis was conducted for $\lambda$, and the results, shown in Table 13, indicate that setting $\lambda = 10$ provides a strong balance of performance across our evaluation metrics.

| Model | #Bits W-A-KV | Method | ARC-c | ARC-e | BoolQ | HellaS. | LambA. | OBQA | PIQA | SIQA | WinoG. | Avg. |
|---|---|---|---|---|---|---|---|---|---|---|---|---|
| | 16-16-16 | Full Precision | 64.42 | 85.98 | 85.14 | 84.95 | 79.47 | 48.46 | 84.39 | 50.82 | 80.66 | 73.81 |
| 3-70B | 4-16-16 | RTN | 26.28 | 25.55 | 37.83 | 26.36 | 0.00 | 29.00 | 50.98 | 34.70 | 49.64 | 31.15 |
| | | SmoothQuant | 51.88 | 77.53 | 80.09 | 80.47 | 73.16 | 46.60 | 80.58 | 45.29 | 75.85 | 67.94 |
| | | GPTQ | 25.77 | 25.29 | 37.83 | 26.36 | 0.12 | 28.40 | 51.74 | 34.90 | 52.64 | 31.45 |
| | | Omniquant | 48.29 | 75.42 | 77.92 | 77.80 | 75.59 | 45.20 | 80.41 | 46.62 | 70.17 | 66.38 |
| | | AWQ | 52.26 | 78.95 | 83.24 | 81.52 | 73.05 | 47.67 | 81.25 | 44.43 | 77.98 | 68.93 |
| | | QuaRot | 62.20 | 83.88 | 85.57 | 84.18 | 79.04 | 48.20 | 83.13 | 50.10 | 80.03 | 72.93 |
| | | SpinQuant | 62.03 | 84.97 | 85.11 | 84.06 | 78.30 | 47.00 | 83.90 | 49.85 | 80.90 | 72.90 |
| | | I$^2$BQ | 63.03 | 85.13 | 84.85 | 84.52 | 79.00 | 48.10 | 83.86 | 50.53 | 80.21 | 73.25 |
| 3-70B | 4-4-16 | RTN | 27.47 | 25.88 | 37.83 | 26.26 | 0.00 | 27.20 | 51.63 | 35.26 | 49.33 | 31.21 |
| | | SmoothQuant | 25.60 | 34.47 | 50.46 | 32.48 | 1.98 | 30.00 | 54.24 | 33.83 | 48.93 | 34.67 |
| | | GPTQ | 25.77 | 26.09 | 43.64 | 26.42 | 0.00 | 27.40 | 52.01 | 32.55 | 49.33 | 31.47 |
| | | QuaRot | 50.60 | 73.65 | 77.46 | 77.83 | 71.96 | 43.20 | 78.13 | 45.29 | 71.90 | 65.56 |
| | | SpinQuant | 53.84 | 77.69 | 80.24 | 78.19 | 73.06 | 45.00 | 78.67 | 43.24 | 73.01 | 66.99 |
| | | I$^2$BQ | 60.49 | 83.99 | 84.01 | 84.21 | 76.69 | 48.30 | 82.69 | 48.81 | 79.33 | 72.09 |
| 3-70B | 4-4-4 | RTN | 27.13 | 25.42 | 37.83 | 26.12 | 0.00 | 26.60 | 50.76 | 35.16 | 48.38 | 30.82 |
| | | SmoothQuant | 23.46 | 31.48 | 48.81 | 29.22 | 4.13 | 28.00 | 52.56 | 34.95 | 51.22 | 33.76 |
| | | GPTQ | 26.11 | 25.17 | 45.17 | 26.07 | 0.00 | 26.40 | 48.86 | 33.88 | 49.17 | 31.20 |
| | | QuaRot | 49.49 | 74.37 | 79.16 | 77.22 | 71.69 | 42.29 | 78.89 | 43.87 | 71.03 | 65.33 |
| | | SpinQuant | 51.88 | 76.39 | 80.98 | 76.50 | 71.43 | 43.46 | 79.27 | 44.17 | 72.69 | 66.31 |
| | | I$^2$BQ | 59.98 | 81.93 | 83.19 | 82.84 | 76.04 | 48.70 | 82.06 | 48.51 | 78.77 | 71.33 |

Table 9: Zero-shot commonsense question answering accuracy (↑) of LLaMA3-70B using different quantization methods and bit-width configurations across multiple datasets.

| Model | #Bits W-A-KV | Method | ARC-c | ARC-e | BoolQ | HellaS. | LambA. | OBQA | PIQA | SIQA | WinoG. | Avg. |
|---|---|---|---|---|---|---|---|---|---|---|---|---|
| | 16-16-16 | Full Precision | 44.71 | 72.90 | 74.98 | 76.20 | 73.08 | 43.80 | 79.16 | 45.55 | 69.93 | 64.48 |
| 7B | 4-16-16 | RTN | 43.17 | 69.82 | 73.30 | 73.75 | 69.67 | 42.00 | 78.13 | 45.34 | 68.82 | 62.67 |
| | | SmoothQuant | 40.96 | 68.60 | 74.04 | 73.16 | 68.74 | 42.00 | 78.07 | 46.11 | 68.51 | 62.24 |
| | | GPTQ | 41.72 | 67.85 | 67.98 | 69.50 | 63.15 | 40.80 | 76.55 | 44.37 | 69.46 | 60.15 |
| | | Omniquant | 42.49 | 71.38 | 74.62 | 74.71 | 71.98 | 42.00 | 79.05 | 45.96 | 68.59 | 63.42 |
| | | AWQ | 43.86 | 70.79 | 74.19 | 75.27 | 69.94 | 43.00 | 78.45 | 45.09 | 69.14 | 63.30 |
| | | QuaRot | 42.75 | 69.99 | 73.30 | 75.13 | 73.55 | 42.00 | 78.35 | 45.14 | 69.61 | 63.40 |
| | | SpinQuant | 43.77 | 71.17 | 74.46 | 75.09 | 72.91 | 44.40 | 78.40 | 44.52 | 70.72 | 63.94 |
| | | I$^2$BQ | 44.17 | 71.92 | 74.38 | 75.07 | 73.37 | 44.40 | 78.17 | 45.69 | 69.65 | 64.09 |
| 7B | 4-4-16 | RTN | 23.46 | 29.34 | 45.05 | 29.02 | 1.24 | 26.00 | 52.07 | 35.11 | 51.30 | 32.51 |
| | | SmoothQuant | 25.17 | 31.40 | 51.62 | 29.73 | 5.43 | 28.20 | 54.68 | 34.44 | 49.09 | 34.42 |
| | | GPTQ | 23.89 | 27.74 | 42.87 | 28.49 | 1.28 | 27.40 | 51.00 | 36.23 | 50.20 | 32.12 |
| | | QuaRot | 40.36 | 67.26 | 73.15 | 72.89 | 70.81 | 42.00 | 77.97 | 44.27 | 67.17 | 61.76 |
| | | SpinQuant | 40.19 | 68.43 | 72.35 | 72.91 | 70.68 | 41.20 | 77.75 | 44.17 | 68.67 | 61.82 |
| | | I$^2$BQ | 41.17 | 69.04 | 73.81 | 72.99 | 71.62 | 42.20 | 78.07 | 44.96 | 68.88 | 62.48 |
| 7B | 4-4-4 | RTN | 23.89 | 29.59 | 46.67 | 28.37 | 1.13 | 26.40 | 52.99 | 35.21 | 51.54 | 32.87 |
| | | SmoothQuant | 23.38 | 30.18 | 50.03 | 29.67 | 4.89 | 24.60 | 51.74 | 34.75 | 50.67 | 33.32 |
| | | GPTQ | 23.89 | 27.90 | 43.88 | 27.86 | 1.05 | 26.20 | 51.85 | 34.08 | 49.49 | 31.80 |
| | | Omniquant | 31.40 | 54.84 | 61.80 | 56.98 | 38.29 | 31.80 | 66.59 | 39.30 | 55.17 | 48.46 |
| | | QuaRot | 40.27 | 67.55 | 72.20 | 72.59 | 70.62 | 39.80 | 77.20 | 44.88 | 65.90 | 61.22 |
| | | SpinQuant | 39.08 | 68.18 | 73.06 | 72.87 | 70.46 | 40.60 | 77.42 | 42.68 | 67.56 | 61.32 |
| | | I$^2$BQ | 41.92 | 69.74 | 73.35 | 72.96 | 71.01 | 41.60 | 77.95 | 43.28 | 67.31 | 62.12 |

Table 10: Zero-shot commonsense question answering accuracy (↑) of LLaMA-7B using different quantization methods and bit-width configurations across multiple datasets.

## A.4 THE USE OF LARGE LANGUAGE MODELS (LLMS)

We used GPT to assist with polishing the writing of this paper. The model was only used to improve grammar, clarity, and readability; all technical content, experiments, and analyses were designed, implemented, and verified by the authors.

| Model | #Bits W-A-KV | Method | ARC-c | ARC-e | BoolQ | HellaS. | LambA. | OBQA | PIQA | SIQA | WinoG. | Avg. |
|---|---|---|---|---|---|---|---|---|---|---|---|---|
| | 16-16-16 | Full Precision | 47.87 | 74.49 | 77.86 | 79.10 | 76.03 | 44.40 | 80.30 | 46.72 | 73.24 | 66.67 |
| 13B | 4-16-16 | RTN | 45.56 | 70.66 | 72.45 | 76.06 | 70.58 | 42.00 | 78.84 | 44.93 | 70.01 | 63.45 |
| | | SmoothQuant | 43.86 | 71.21 | 71.62 | 74.19 | 69.34 | 40.00 | 77.80 | 45.45 | 70.72 | 62.69 |
| | | GPTQ | 45.99 | 72.85 | 73.27 | 75.31 | 70.10 | 44.60 | 79.87 | 46.16 | 71.11 | 64.36 |
| | | Omniquant | 47.01 | 73.86 | 77.22 | 77.95 | 75.59 | 45.00 | 79.87 | 46.88 | 72.61 | 66.22 |
| | | AWQ | 47.53 | 73.86 | 75.60 | 59.03 | 78.34 | 43.40 | 79.87 | 45.85 | 71.67 | 65.58 |
| | | QuaRot | 47.18 | 72.22 | 76.85 | 78.07 | 75.99 | 45.00 | 79.76 | 45.70 | 72.38 | 65.91 |
| | | SpinQuant | 47.44 | 74.83 | 77.37 | 78.13 | 75.55 | 45.60 | 79.92 | 46.01 | 72.06 | 66.32 |
| | | I$^2$BQ | 47.41 | 74.72 | 77.72 | 78.08 | 75.92 | 45.70 | 80.22 | 46.15 | 72.31 | 66.47 |
| 13B | 4-4-16 | RTN | 25.85 | 26.26 | 42.05 | 26.70 | 0.17 | 28.00 | 50.33 | 34.60 | 50.67 | 31.63 |
| | | SmoothQuant | 25.43 | 29.29 | 51.56 | 28.12 | 2.02 | 26.00 | 53.32 | 34.34 | 49.57 | 33.29 |
| | | GPTQ | 24.66 | 27.78 | 40.80 | 25.83 | 0.70 | 24.20 | 51.31 | 36.65 | 51.70 | 31.51 |
| | | QuaRot | 46.93 | 71.51 | 75.57 | 76.63 | 74.13 | 42.40 | 78.73 | 45.24 | 68.98 | 64.46 |
| | | SpinQuant | 45.73 | 72.56 | 75.38 | 76.86 | 73.28 | 43.60 | 78.89 | 44.63 | 70.40 | 64.59 |
| | | I$^2$BQ | 47.38 | 73.71 | 77.22 | 76.88 | 74.66 | 44.60 | 78.86 | 45.67 | 71.03 | 65.56 |
| 13B | 4-4-4 | RTN | 26.28 | 27.27 | 42.35 | 25.85 | 0.19 | 26.60 | 49.95 | 34.19 | 49.25 | 31.33 |
| | | SmoothQuant | 24.49 | 28.83 | 51.65 | 27.91 | 2.08 | 26.00 | 52.56 | 35.41 | 50.59 | 33.28 |
| | | GPTQ | 23.63 | 27.31 | 39.85 | 26.17 | 0.56 | 26.00 | 51.96 | 35.82 | 49.57 | 30.63 |
| | | Omniquant | 29.61 | 48.23 | 58.20 | 56.45 | 28.76 | 31.40 | 65.29 | 37.10 | 55.64 | 45.63 |
| | | QuaRot | 46.50 | 71.55 | 75.08 | 76.43 | 73.47 | 45.00 | 78.78 | 44.37 | 70.09 | 64.59 |
| | | SpinQuant | 45.99 | 70.71 | 76.51 | 77.16 | 73.63 | 45.60 | 79.00 | 45.65 | 70.32 | 64.95 |
| | | I$^2$BQ | 46.02 | 73.23 | 77.09 | 76.57 | 74.07 | 45.30 | 78.64 | 45.77 | 70.38 | 65.21 |

Table 11: Zero-shot commonsense question answering accuracy (↑) of LLaMA-13B using different quantization methods and bit-width configurations across multiple datasets.

| MWO | CBEC | BWQ | WikiText-2(↓) | C4(↓) | ARC-C | ARC-E | HellaSwag | LAMBADA | PIQA | Winogrande | Avg(↑) |
|---|---|---|---|---|---|---|---|---|---|---|---|
| | | | 20.11 | 21.02 | 23.89 | 52.53 | 36.60 | 60.18 | 64.53 | 55.09 | 41.74 |
| | | ✓ | 7.01 | 8.58 | 36.43 | 68.73 | 52.75 | 57.25 | 74.43 | 63.46 | 51.78 |
| ✓ | | | 6.23 | 7.87 | 40.53 | 73.48 | 53.86 | 66.63 | 76.17 | 65.19 | 58.84 |
| ✓ | ✓ | | 5.96 | 7.52 | 40.87 | 74.07 | 54.81 | 67.07 | 76.06 | 67.01 | 63.32 |

Table 12: Ablation study of the main components of I$^2$BQ on LLaMA-2-7B under the W4A4 setting. ↓ is better for perplexity (WikiText-2, C4), while ↑ is better for downstream task accuracy.

| λ | ARC-c | ARC-e | BoolQ | HellaS. | Lam. | OBQA | PIQA | SIQA | WinoG. | Avg. (↑) |
|---|---|---|---|---|---|---|---|---|---|---|
| 0.1 | 40.97 | 72.59 | 73.71 | 74.23 | 69.50 | 43.53 | 76.21 | 44.80 | 65.14 | 62.30 |
| 1 | 39.29 | 73.28 | 74.52 | 73.87 | 70.87 | 43.15 | 77.03 | 43.49 | 66.57 | 62.45 |
| 10 | 40.87 | 74.07 | 74.89 | 74.81 | 70.67 | 43.89 | 76.06 | 44.79 | 67.01 | 63.00 |
| 15 | 41.04 | 71.88 | 72.61 | 74.26 | 68.96 | 44.16 | 76.85 | 44.00 | 67.42 | 62.35 |
| 20 | 40.63 | 72.67 | 73.02 | 73.85 | 69.40 | 44.07 | 77.21 | 43.40 | 65.99 | 62.24 |

Table 13: Sensitivity analysis of the coefficient λ on zero-shot accuracy (↑) across multiple benchmarks.

