# OpenReview forum: "I$^2$BQ: Quantizing LLMs via Intra- and Inter-Block Optimization"
_ICLR.cc/2026/Conference — ICLR 2026 Conference Withdrawn Submission_

### Official Review · Reviewer_Le7b · 2025-10-18

**Soundness:** 2
**Presentation:** 2
**Contribution:** 2
**Rating:** 4
**Confidence:** 5

**Summary:**

In this work, a new post-training quantization algorithm for LLMs has been proposed.

The key contributions include
 - separate consideration of self-attention modules and MLP layers
 - inclusion of a new loss term that aims to match the attention score distributions of quantized and original models
 - consideration of cross-block dependencies.

Through experiments on publicly available LLMs (e.g., OPT and Llama families), the effectiveness of the proposed method has been validated.

**Strengths:**

- The idea to consider self-attention modules and MLP layers separately is well-motivated.
- To my knowledge, the ideas to 1) match the attention score distributions of quantized and full-precision models and 2) consider cross-block dependencies using the loss function in (8) are novel.

**Weaknesses:**

- Except for the contributions that I mentioned in Strengths, other contributions are marginal and can be found in existing PTQ works.
- Recent baselines are not included in the comparison.
- Some statements need to be clarified further.

For more details, please refer to my comments in Questions.

**Questions:**

0. Processing time comparison and more baselines
 - The most crucial concern is the long processing time, as the authors pointed out as their limitation.
 - Computing the loss values in Eq. (8) would be very time-consuming and computationally heavy, since we need to conduct the forward operations of multiple blocks in each quantization iteration.
 - The authors need to report the quantization processing time.
 - Moreover, recent baselines such as aespa and GPTAQ have not been compared. Aespa also considers the attention module separately, and GPTAQ also tries to reduce the error propagation, so comparison with those methods are crucial. Please compare with them using small-scale models such as OPT-125M, Llama3.2-1b-instruct, Llama3.2-3b-instruct without the rotation to solely compare the performance.

1. About the separate consideration of each module
 - To my knowledge, considering the entire Transformer block is more beneficial, because dependencies between more layers can be considered.
 - The authors claimed that the module-wise optimization (MWO) performs better than the block-wise quantization error minimization (BWQ) by providing the empirical results in Table 4. However,  when applying BRECQ to quantize OPT-125M by myself, I observed that setting the block unit as the entire Transformer block performs better than quantizing self-attention and MLP modules separately. Could the authors give more justification?

2. About the loss function
 - How did the authors combine the loss functions in Eqs. (6), (7), and (8)?
 - Why did the authors match the attention scores rather than matching the product of attention scores and value outputs?
 - What are the trainable parameters? Do the authors train only clipping coefficients in Eq. (2) or learn additional parameters related to the rounding as in AutoRound or BRECQ?

3. About the figures
 - Which activations did the authors plot in Figures 2 and 3? Input activations for each module?
 - How did the authors illustrate the Hessian matrix for all the parameters inside self-attention modules in Figure 4? There are a huge number of parameters in Llama2-7B, so I think computing the Hessian matrix is challenging.

4. About the experimental results
 - Did the authors measure the performance of conventional methods by themselves under the same calibration dataset? Or just report the results?
 - I want to see the low-bit result for Llama3 models (e.g., Llama3.2-1b-instruct, Llama3.2-3b-instruct), which are more senstive to quantization.

---

### Official Review · Reviewer_mAGT · 2025-10-29

**Soundness:** 2
**Presentation:** 2
**Contribution:** 2
**Rating:** 2
**Confidence:** 5

**Summary:**

The paper builds upon the OmniQuant framework and introduces several quantization strategies motivated by heuristic observations. Specifically, it addresses the issue of cross-layer quantization error propagation by varying the quantization granularity across layers, and further establishes module-specific optimization objectives by decoupling the self-attention and feedforward modules.

**Strengths:**

- The paper extends OmniQuant by introducing a new quantization strategy that enhances overall performance. The proposed approach is designed based on empirical observations to account for intra- and inter-block dependencies within Transformer architectures, aiming to minimize the propagation of accumulated quantization errors during the quantization process.

- the authors introduce an attention-preserving loss that quantitatively reflects inter-token dependencies, thereby distinguishing their method from prior works and providing a more dependency-aware quantization perspective.

**Weaknesses:**

- The paper lacks an ablation study on the proposed attention-preserving loss. It would be also beneficial to include experiments analyzing the effect of the λ (lambda) weighting parameter on performance.

- The number of experimental runs is not specified. Repeated trials and evaluations on multiple datasets are necessary to demonstrate the method’s robustness and statistical reliability.

-The comparison is limited to outdated models, and the improvements over OmniQuant appear incremental and marginal, which weakens the perceived novelty. Since the approach is primarily based on heuristic observations, additional experiments on diverse and recent architectures (e.g., Gemma, Qwen) would strengthen the generality claim.

- The baseline algorithms used for comparison are also outdated. Comparative evaluations against recent quantization methods such as BoA (https://arxiv.org/abs/2406.13474), aespa (https://arxiv.org/abs/2402.08958), and FlatQuant (https://arxiv.org/abs/2410.09426), along with both qualitative and quantitative analyses, are required to clarify the advantages and distinctions of the proposed method.

**Questions:**

see weaknesses

---

### Official Review · Reviewer_Zbvg · 2025-10-31

**Soundness:** 3
**Presentation:** 1
**Contribution:** 2
**Rating:** 2
**Confidence:** 3

**Summary:**

This paper introduces I2BQ, a quantization framework that separately quantizes attention and FFN modules, and alleviates inter-layer error accumulation by jointly optimizing multiple blocks.

**Strengths:**

* Error accumulation across layers remains an important challenge in LLM quantization.

* The idea of performing quantization in a more fine-grained manner than conventional block-wise approaches is novel, and the underlying intuition appears well-motivated.

* The paper includes extensive evaluations.

**Weaknesses:**

* Incomplete explanation of cross-block error compensation.

The paper’s description of cross-block compensation is unclear. It is not specified whether error compensation is performed after module-wise quantization via a separate optimization step, or together with the module-wise optimization in a joint procedure. If it is joint, essential details on  how the optimization is formulated (objective/loss) are missing. The sentence “in practice, this loss can be applied within a specific module (e.g., self-attention or FFN) by computing the reconstruction error from the module in block i to the corresponding module in block i+n” seems to explain this process but it sounds unclear to me.

And cross-block error compensation seems to introduce hyper-parameter $n$ which denotes how many blocks would be considered together, but the paper does not discuss how to set $n$ and which value is used for evaluations.

* * Discussion/comparison with closely related work is missing.

Cross-block error compensation is an important pillar of this paper, but there have been multiple proposals that address this issue although with some different approaches. First of all, as authors include in the evaluations, CBQ is the one. But there is only accuracy comparison, and no qualtitative discussion how the proposed approach is different from CBQ and why it would be a better approach. Also, there are some approaches that reduce inter-layer error accumulation by fine-tuning like in AQLM [1] and QUIP# [2], and I think the authors should include discussion on this line of works.

[1] Egiazarian et al., "Extreme Compression of Large Language Models via Additive Quantization", ICML'24

[2] Tseng et al., "QuIP#: Even Better LLM Quantization with Hadamard Incoherence and Lattice Codebooks", ICML'24

**Questions:**

Could you please clarify the two points I raised in the weakness section?
Also, I am curious about the quantization cost like how long the quantization process takes and on which device it was performed.

---

### Official Review · Reviewer_Qj8A · 2025-10-31

**Soundness:** 1
**Presentation:** 1
**Contribution:** 1
**Rating:** 2
**Confidence:** 4

**Summary:**

The paper treats attention and FFN layers as separate distribution sources which therefore warrant differing quantization strategies. This essentially means that the attention part is trainable, whereas FFNs are frozen. They also propose cross-block quantization error corrections.

**Strengths:**

- Nothing in particular.

**Weaknesses:**

- There is no theoretical or analytical foundation to the work. The contributions are arbitrary and ad-hoc. This does not advance the state-of-the-art and is unfit for a conference like ICLR.
- I2BQ does not lead to a better pareto frontier. For instance, OPT-66B 2b has roughly the same size as OPT-30B 4b. The former has wiki PPL of 9.82 which is worse than the latter which has wiki PPL of 9.59. Thus rather than using I2BQ and quantizing to 2b, it would have been better to stick to 4 bits and use a twice smaller model.

**Questions:**

N/A

---

> ### Comment · Reviewer_Qj8A · 2025-11-25
> **No rebuttal**
>
> Since there is no rebuttal, I maintain my recommendation to reject this paper.

---

### Note · Authors · 2026-01-06

I have read and agree with the venue's withdrawal policy on behalf of myself and my co-authors.